# D³Fields: Dynamic 3D Descriptor Fields for Zero-Shot Generalizable Rearrangement

Yixuan Wang[1*], Mingtong Zhang[2*], Zhuoran Li[3*], Tarik Kelestemur[4],
Katherine Driggs-Campbell[2], Jiajun Wu[5], Li Fei-Fei[5], Yunzhu Li[1]

[1]Columbia University, [2]University of Illinois, Urbana-Champaign, [3]National University of Singapore,
[4]Boston Dynamics AI Institute, [5]Stanford University

**Abstract:** Scene representation is a crucial design choice in robotic manipulation systems. An ideal representation is expected to be 3D, dynamic, and semantic to meet the demands of diverse manipulation tasks. However, previous works often lack all three properties simultaneously. In this work, we introduce D³Fields—**dynamic 3D descriptor fields**. These fields are **implicit 3D representations** that take in 3D points and output semantic features and instance masks. They can also capture the dynamics of the underlying 3D environments. Specifically, we project arbitrary 3D points in the workspace onto multi-view 2D visual observations and interpolate features derived from visual foundational models. The resulting fused descriptor fields allow for flexible goal specifications using 2D images with varied contexts, styles, and instances. To evaluate the effectiveness of these descriptor fields, we apply our representation to rearrangement tasks in a zero-shot manner. Through extensive evaluation in real worlds and simulations, we demonstrate that D³Fields are effective for **zero-shot generalizable** rearrangement tasks. We also compare D³Fields with state-of-the-art implicit 3D representations and show significant improvements in effectiveness and efficiency. Project Page

**Keywords:** Implicit 3D Representation, Visual Foundational Model, Zero-Shot Generalization, Robotic Manipulation

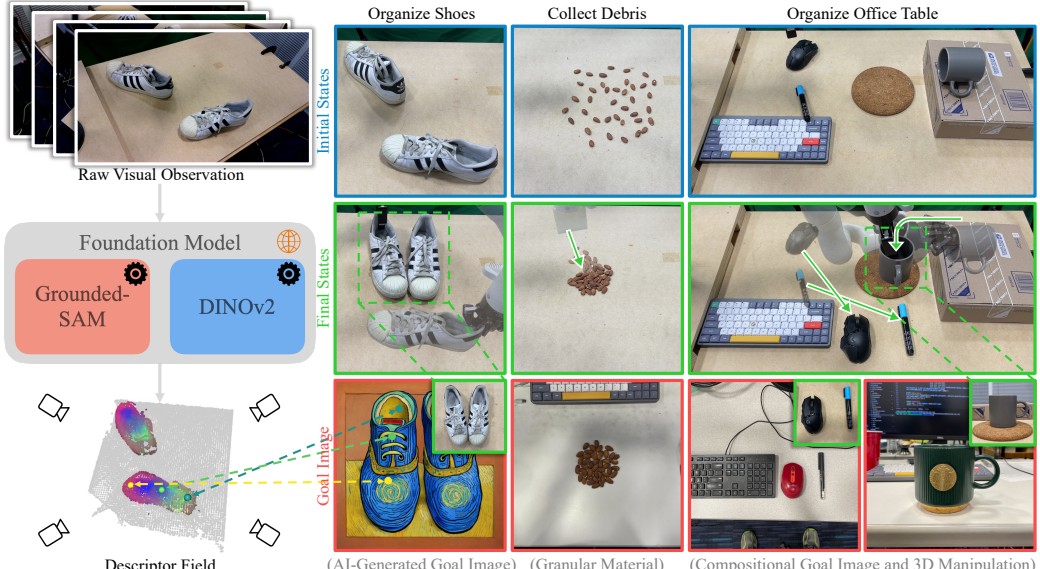

Figure 1: **D³Fields Representation and Application to Zero-Shot Rearrangement Tasks.** D³Fields take in multi-view RGBD images and encode semantic features and instance masks using foundational models. The descriptor fields visualized in the bottom left using Principal Component Analysis (PCA) demonstrate consistent features across instances. We use our representation for rearrangement tasks given 2D goal images with diverse instances and styles in a zero-shot manner. We address pick-and-place tasks such as shoe organization and tasks requiring dynamic modeling like collecting debris. We also show that our framework can accomplish 3D manipulation and compositional task specification in the table organization task.

* Denotes equal contribution.
8th Conference on Robot Learning (CoRL 2024), Munich, Germany.

# 1 Introduction

The choice of scene representation is essential in robotic systems. An ideal representation is expected to be simultaneously 3D, dynamic, and semantic to meet the needs of various robotic manipulation tasks in our daily lives. However, previous research on scene representations in robotics often does not encompass all three properties. Some representations exist in 3D space [1–4], yet they overlook semantic information. Others focus on dynamic modeling [5–8], but only consider 2D data, neglecting the role of 3D space. Some other works are limited by only considering semantic information such as object instance and category [9–13].

In this work, we aim to satisfy all three criteria by introducing $D^3$Fields, unified descriptor fields that are **3D, dynamic, and semantic**. Notably, $D^3$Fields are **implicit 3D representations** rather than explicit 3D representations like point clouds. $D^3$Fields take arbitrary 3D coordinates as inputs and output both geometric and semantic information corresponding to these positions. This includes the instance mask, dense semantic features, and the signed distance to the object surface. Notably, deriving these descriptor fields requires no training and is conducted in a zero-shot manner, utilizing large visual foundation models and vision-language models (VLMs). In our approach, we employ a set of advanced models. We first use Grounding-DINO [14], Segment Anything (SAM) [15], XMem [16], and DINOv2 [17] to extract information from multi-view 2D RGB images. We then project arbitrary 3D coordinates back to each camera, interpolate to compute representations from each view, and fuse these data to derive the descriptors associated with these 3D positions, as shown in Figure 1 (left). Leveraging the dense semantic features and instance mask of our representation, we achieve robust tracking 3D points of the target object instances and train the dynamics models. These learned dynamics models can be incorporated into a Model-Predictive Control (MPC) framework to plan for zero-shot generalizable rearrangement tasks.

Notably, the derived representations allow for zero-shot generalizable rearrangement tasks, where the goal is specified by 2D images sourced from the Internet, smartphones, or even generated by AI models. Such goal images have been challenging to manage with previous methods, because they contain varied styles, contexts, and object instances different from the robot's workspace. Our proposed $D^3$Fields can establish dense correspondences between the robot workspace and the target configurations. Given correspondences, we can define our planning cost and use the MPC framework with the learned dynamics model to derive actions for accomplishing tasks. Remarkably, this task execution process does not require any further training, offering a highly flexible and convenient interface for humans to specify tasks for the robots.

We evaluate our method across a wide range of robotic rearrangement tasks in a zero-shot manner. These tasks include organizing shoes, collecting debris, and organizing office desks, as shown in Figure 1 (right). Furthermore, we provide both quantitative and qualitative comparisons with state-of-the-art implicit 3D representations to demonstrate the effectiveness and efficiency of our approach [18, 19]. Through a detailed analysis of our $D^3$Fields, we offer insights into the category-level generalization capabilities and zero-shot rearrangement capabilities of our approach.

We make three major contributions. First, we introduce a novel representation, $D^3$Fields, that is **3D**, **semantic**, and **dynamic**. Second, we present a novel and flexible goal specification method using 2D images that incorporate a wide range of styles, contexts, and instances. Third, our proposed robotic manipulation framework supports a broad spectrum of zero-shot rearrangement tasks.

# 2 Related Works

**Foundation Models for Robotics**. Large Language Models (LLMs) have demonstrated promising reasoning capabilities for language. Robotics researchers have used LLMs to generate plans for manipulation [20–23]. Yet, their perception modules fall short in simultaneously modeling the 3D geometry, semantics, and dynamics of objects. Meanwhile, visual foundation models, such as SAM [15] and DINOv2 [17], have demonstrated impressive zero-shot generalization capabilities across various vision tasks. While prior visual models, like Dense Object Nets [24], can encode similar semantic information on a small-scale dataset, these foundational models show better gen-

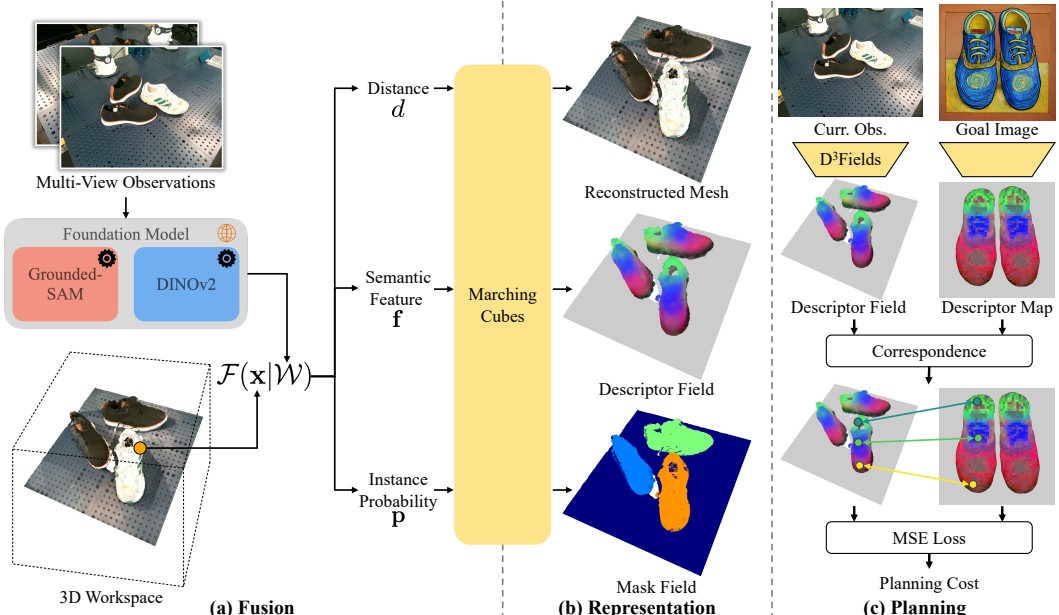

Figure 2: **Overview of the Proposed Framework.** (a) Multi-view RGBD observations are first processed by foundation models to obtain the feature volume $\mathcal{W}$ The implicit function $\mathcal{F}$ takes in arbitrary 3D points and outputs corresponding distance $d$, semantic features $\mathbf{f}$, and instance probability $\mathbf{p}$. (b) Through marching cubes, we could reconstruct the mesh from the implicit signed distance function. Since our representation also encodes instances masks and semantic features for evaluated 3D points, we can construct meshes for the mask field and descriptor field as well. (c) Given a 2D goal image, we use foundation models to extract the descriptor map. Then we correspond 3D features to 2D features and define the planning cost based on the correspondence.

eralization capabilities on various object categories and scenarios. However, their focus is primarily on 2D vision tasks. Grounding these models in a dynamic 3D environment remains a challenge. Recent works showcase how to ground these foundational models in the 3D world and help imitation learning to generalize [25–28]. Still, these works do not emphasize dynamics learning or achieve zero-shot generalization ability.

**Neural Fields for Robotic Manipulation.** There are various approaches leveraging neural fields as a representation for robotic manipulation [29–42]. Among them, a series of works distilling neural feature fields from visual foundation models are closely related to us [19, 43–46]. However, they often require dense camera views for a quality field, which is expensive and impractical for real-world scenarios. Also, distilled neural fields need retraining for new scenes, which is time-consuming and inefficient. In contrast, our D³Fields require no training for new scenes and can work with sparse views and dynamic settings. GNFactor and FeatureNeRF train neural feature fields that can be conditioned on sparse views [18, 47]. However, such fields are often trained on a small dataset, making them hard to generalize to novel instances and scenes, whereas our D³Fields offer better generalization capability to new instances.

## 3 Method

### 3.1 Problem Formulation

We formulate our problem as a zero-shot rearrangement problem given a 2D goal image $\mathcal{I}$ and RGBD images from multiple fixed viewpoints. We denote the workspace scene representation as $\boldsymbol{s}_{\text{goal}}$. Our goal is to find an optimal action sequence $\{a^t\}$ to minimize the task objective:

$$
\begin{aligned}
\min_{\{a_t\}} \quad & c(\boldsymbol{s}^T, \boldsymbol{s}_{\text{goal}}), \\
\text{s.t.} \quad & \boldsymbol{s}^t = g(\boldsymbol{o}^t), \quad \boldsymbol{s}^{t+1} = f(\boldsymbol{s}^t, a^t),
\end{aligned}
\tag{1}
$$

where $c(\cdot, \cdot)$ is the cost function measuring the distance between the terminal representation $\boldsymbol{s}^T$ and the goal representation $\boldsymbol{s}_{\text{goal}}$. Representation extraction function $g(\cdot)$ takes in the current multi-view

RGBD observations $\boldsymbol{o}^t$ and outputs the current representation $\boldsymbol{s}^t$. $f(\cdot, \cdot)$ is the dynamics function that predicts the future representation $\boldsymbol{s}^{t+1}$, conditioned on the current representation $\boldsymbol{s}^t$ and action $a^t$. The optimization aims to find the action sequence $\{a_t\}$ that minimizes the cost function $c(\boldsymbol{s}^T, \boldsymbol{s}_{\text{goal}})$.

Section 3.2 describes how to construct representation extraction function $g(\cdot)$. Section 3.3 elaborates on the dynamics function $f(\cdot, \cdot)$. In Section 3.4, we show how to define the cost function $c(\cdot, \cdot)$.

### 3.2 D³Fields Representation

We assume that we can access multiple RGBD cameras with fixed viewpoints to construct D³Fields. Multi-view RGBD observations are first fed into visual foundational models. Then we obtain 2D feature volumes $\mathcal{W}$. D³Fields are implicit functions $\mathcal{F}(\cdot | \mathcal{W})$ defined as follows:

$$(d, \mathbf{f}, \mathbf{p}) = \mathcal{F}(\mathbf{x} | \mathcal{W}), \tag{2}$$

where $\mathbf{x}$ can be an arbitrary 3D coordinate in the world frame, and $(d, \mathbf{f}, \mathbf{p})$ corresponds to the signed distance $d \in \mathbb{R}$, the semantic descriptor $\mathbf{f} \in \mathbb{R}^N$, and the instance probability distribution $\mathbf{p} \in \mathbb{R}^M$ of $M$ instances. $M$ could be different across scenarios.

As an overview, our pipeline first projects $\mathbf{x}$ into the image space of each camera. Then, we can obtain the truncated depth difference between projected depth and real depth reading using Equation 3. Afterwards, we assign weights to each viewpoint using Equation 4 and interpolate the semantic features and instance masks for each camera using Equation 5. Finally, we fuse features from all viewpoints to obtain the final descriptor using Equation 6.

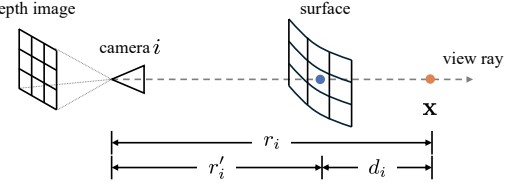

Figure 3: **Notation Illustration.** $r_i$ is the distance between a 3D point $\mathbf{x}$ and camera $i$, and $r_i'$ is the interpolated depth from the depth image.

More concretely, we map an arbitrary 3D point $\mathbf{x}$ to the $i$th viewpoint's image space. We denote the projected pixel as $\mathbf{u}_i$ and the distance from $\mathbf{x}$ to the $i$th viewpoint as $r_i$ (Figure 3). By interpolating the $i$th viewpoint's depth image $\mathcal{R}_i$, we compute the corresponding depth reading from the depth image as $r_i' = \mathcal{R}_i[\mathbf{u}_i]$. Then we can compute the truncated depth difference as

$$d_i = r_i - r_i', \quad d_i' = \max(\min(d_i, \mu), -\mu), \tag{3}$$

where $\mu$ specifies the truncation threshold for the Truncated Signed Distance Function (TSDF). Given the truncated depth difference, we compute weights $v_i$ and $w_i$ for each viewpoint as

$$v_i = \mathbb{1}_{d_i < \mu}, \quad w_i = \exp\left(\frac{\min(\mu - |d_i|, 0)}{\mu}\right). \tag{4}$$

Here is the explanation and design justification for each term.

$v_i$: It represents the visibility of $\mathbf{x}$ in camera $i$. $\mathbb{1}_{d_i < \mu}$ is the indicator function, which equals to 1 when $d_i < \mu$ and equals to 0 otherwise. When $d_i = r_i - r_i' \geq \mu$, $\mathbf{x}$ is behind the surface, which means $\mathbf{x}$ is not visible in camera $i$ and $v_i = 0$.

$w_i$: It is the weight for the $i$th viewpoint. Since we only have a confident estimation when $\mathbf{x}$ is close to the surface, $w_i$ will decay as $|d_i|$ increases. For $\mathbf{x}$ that is far away, $w_i$ degrades to 0.

Then we extract the semantic feature $\mathbf{f}_i$ and instance mask $\mathbf{p}_i$ in each viewpoint using

$$\mathbf{f}_i = \mathcal{W}_i^{\mathbf{f}}[\mathbf{u}_i], \quad \mathbf{p}_i = \mathcal{W}_i^{\mathbf{p}}[\mathbf{u}_i], \tag{5}$$

where DINOv2 [17] extracts the semantic feature volume $\mathcal{W}_i^{\mathbf{f}} \in \mathbb{R}^{H \times W \times N}$ from RGB image. $\mathcal{W}_i^{\mathbf{P}} \in \mathbb{R}^{H \times W \times M}$ is the instance mask volume using Grounded-SAM [14, 15]. Note that $\boldsymbol{p}_i$ is a one-hot vector and already associated to ensure consistent instance indexing across different views. Finally, we fuse the semantic features and instance masks from all $K$ viewpoints using

$$d = \frac{\sum_{i=1}^{K} v_i d_i'}{\delta + \sum_{i=1}^{K} v_i}, \quad \mathbf{f} = \frac{\sum_{i=1}^{K} v_i w_i \mathbf{f}_i}{\delta + \sum_{i=1}^{K} v_i}, \quad \mathbf{p} = \frac{\sum_{i=1}^{K} v_i w_i \mathbf{p}_i}{\delta + \sum_{i=1}^{K} v_i}, \tag{6}$$

where $\delta$ is a small number to avoid numeric issues. Since the process of projection, interpolation, and fusion is differentiable, $\mathcal{F}(\cdot | \mathcal{W})$ is differentiable when $\mathbf{x}$ is within the truncation threshold.

### 3.3 Keypoints Tracking and Dynamics Learning

This section will present how to use the dynamic implicit 3D descriptor field $\mathcal{F}(\cdot|\mathcal{W})$ to track keypoints and train dynamics. Without loss of generality, consider the tracking of a single object instance $\boldsymbol{s}^t \in \mathbb{R}^{3 \times n_s}$. For clarity, we denote $\mathbf{f}$ and $d$ from $\mathcal{F}(\cdot|\mathcal{W})$ as $\mathcal{F}_{\mathbf{f}}(\cdot|\mathcal{W})$ and $\mathcal{F}_d(\cdot|\mathcal{W})$. We initialize the tracked keypoints $\boldsymbol{s}^0$ by sampling points close to the surface of the desired instance. To track keypoints $\boldsymbol{s}^0$, we formulate the tracking problem as an optimization problem:

$$\min_{\boldsymbol{s}^{t+1}} \quad ||\mathcal{F}_{\mathbf{f}}(\boldsymbol{s}^{t+1}|\mathcal{W}) - \mathcal{F}_{\mathbf{f}}(\boldsymbol{s}^0|\mathcal{W})||_2. \tag{7}$$

As $\mathcal{F}(\cdot|\mathcal{W})$ is differentiable, we could use a gradient-based optimizer. This method could be naturally extended to multiple-instance scenarios. We found that relying solely on features for tracking can be unstable. Therefore, if we know that the tracked object is rigid, we can apply additional rigid constraints and distance regularization for more stable tracking.

Keypoint tracking enables dynamics model training on real data. We instantiate the dynamics model $f(\cdot, \cdot)$ as graph neural networks (GNNs). We follow [48] to predict object dynamics. Please refer to [48, 49] for more details on how to train the graph-based neural dynamics model. The trained dynamics model will be used for trajectory optimization in Section 3.4.

### 3.4 Zero-Shot Generalizable Robotic Rearrangement

In this section, we will describe how to define the planning cost for our zero-shot rearrangement framework. As shown in Figure 2 (c), we first find the correspondence between the descriptor fields and goal image using Equation 8. Then we define the cost function $c(\cdot, \cdot)$ in Equation 9 to measure the distance between the current state and the goal state. Finally, we optimize the action sequence $\{a_t\}$ to minimize the cost function as described in Section 3.1.

As described in Section 3.2, we initially sample points $\boldsymbol{s}^0 \in \mathbb{R}^{3 \times n_s}$ and obtain the associated features $\mathbf{f}^0 \in \mathbb{R}^{f \times n_s}$ from the descriptor fields. We correspond $\boldsymbol{s}^0$ to the goal image $\mathcal{I}_{\text{goal}}$ to define 2D goal points $\boldsymbol{s}_{\text{goal}} \in \mathbb{R}^{2 \times n_s}$. Firstly, we compute the feature distance $\alpha_{ij}$ between $i$th pixel $\mathbf{u}_i$ of $\mathcal{I}_{\text{goal}}$ and $j$th sampled point of $\boldsymbol{s}^0$. Then we normalize $\alpha_{ij}$ using the softmax over the whole image and obtain the weight $\beta_{ij}$. Lastly, we find the 2D point $\boldsymbol{s}_{\text{goal},j}$ corresponding to the $j$th 3D point using weighted sum. The computation process is summarized in the following:

$$\alpha_{ij} = ||\mathcal{W}_{\text{goal}}^{\mathbf{f}}[\mathbf{u}_i] - \mathbf{f}_j^0||_2, \quad \beta_{ij} = \frac{\exp(-s\alpha_{ij})}{\sum_{i=1}^{H \times W} \exp(-s\alpha_{ij})}, \quad \boldsymbol{s}_{\text{goal},j} = \sum_{i=1}^{H \times W} \beta_{ij}\mathbf{u}_i, \tag{8}$$

where $\mathcal{W}_{\text{goal}}^{\mathbf{f}}$ is the feature volume extracted from $\mathcal{I}_{\text{goal}}$ using DINOv2, and $s$ is the hyperparameter to determine whether the heatmap $\beta_{ij}$ is more smooth or concentrating. Although Equation. 8 only shows a single instance case, it could be naturally extended to multiple instances by using instance mask information.

Note that $\boldsymbol{s}_{\text{goal}}$ is in the image space, and the current state representation, $\boldsymbol{s}^t$ is in 3D space. To reconcile this discrepancy, we introduce a *virtual* reference camera. We project $\boldsymbol{s}^t$ into the reference image and obtain its 2D positions $\boldsymbol{s}_{\text{2D}}^t$. Consequently, we define the task cost function as follows:

$$c(\boldsymbol{s}^t, \boldsymbol{s}_{\text{goal}}) = ||\boldsymbol{s}_{\text{2D}}^t - \boldsymbol{s}_{\text{goal}}||_2^2. \tag{9}$$

Given the planning cost, we could use an MPC framework to derive the action sequence $\{a_t\}$ to minimize the cost function. Specifically, we use MPPI to optimize the action sequence [50]. At each time step, we sample a set of action sequences and evaluate the cost function. Then we update the action sequence using the cost function and repeat the process until conver-

Figure 4: **Object Set in Our Experiments.** This figure shows diverse objects used in our experiments, expanding over 10 object types.

gence. This process will be repeated for each time step until the task is completed. More details regarding the method are included in the supplementary material.

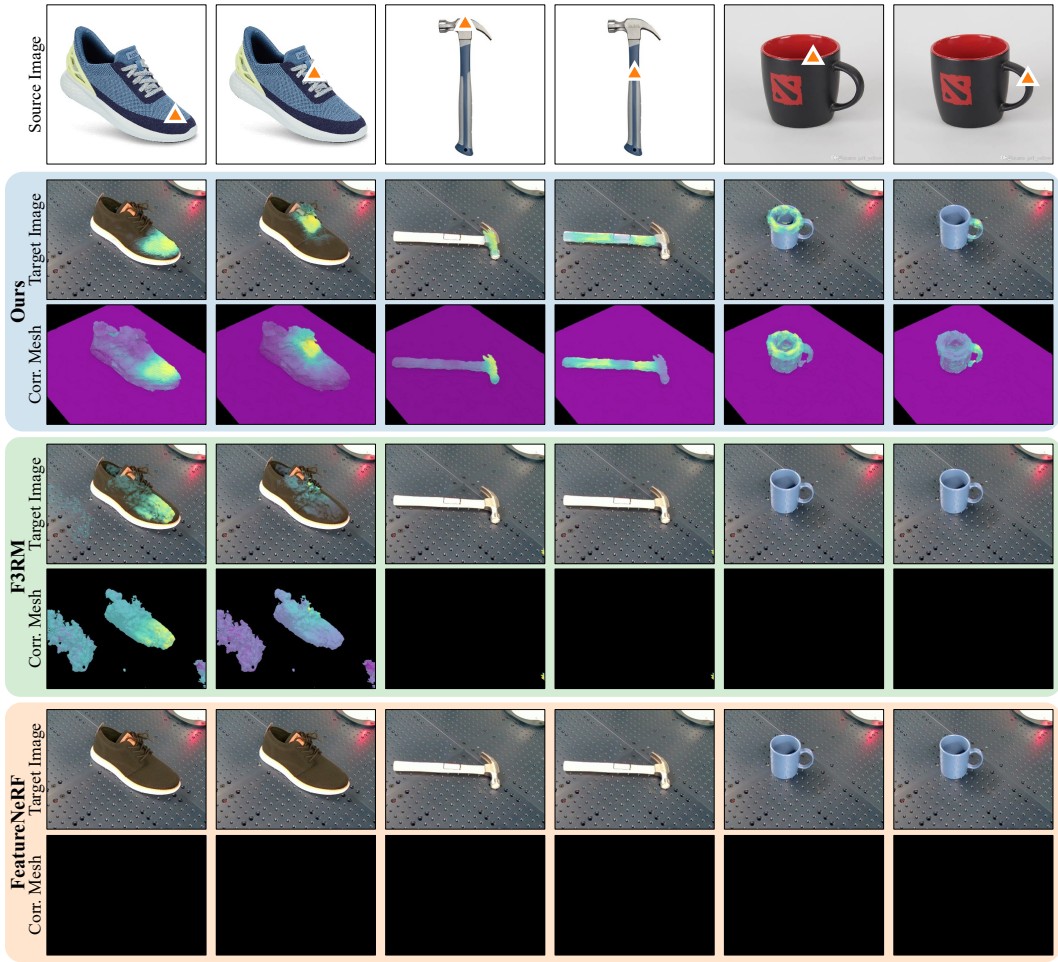

Figure 5: **Correspondence Qualitative Comparison.** We select the pixel from the source image, obtain the associated DINOv2 feature, and visualize the correspondence heatmap on the reconstructed mesh. (a) Our representation reconstructs clear mesh and corresponds from the source image to semantically similar 3D areas. (b) F3RM [19] could construct a reasonable mesh in the shoe scene and establish rough correspondences, but fails in other scenes. (c) Only trained on a small dataset, FeatureNeRF [18] fails to generalize to novel scenes. The reconstructed meshes are out of camera view, and the correspondence quality is poor.

## 4 Experiments

We design and organize our experiments to answer three key questions about our method: (1) How efficient and effective our D³Fields is compared to existing neural representations? (2) What kind of manipulation tasks can be enabled by our framework, and what type of generalization can it achieve? (3) Why can our D³Fields enable these tasks and be generalizable?

### 4.1 Experiment Setup

In the real world, we use four RGBD cameras positioned at the corners of the workspace and employ the Kinova® Gen3 arm for action execution. In the simulation, we use OmniGibson and the Fetch robot for mobile manipulation tasks [51]. Our evaluations span various tasks, including organizing shoes, collecting debris, tidying the table, and so on. More details are in the supplementary material.

### 4.2 D³Fields Efficiency and Effectiveness

In this section, we compare our D³Fields with two baselines, Distilled Feature Fields (F3RM) [19] and FeatureNeRF [18]. For F3RM, we use four camera views as inputs and the DINOv2 features as the supervision and stop the distillation process at 2,000 steps, which is defaulted by the authors [19]. We first compare our D³Fields with F3RM and FeatureNeRF in terms of the correspondence accuracy, as shown in Figure 5. We reconstruct the mesh from our 3D implicit representation using marching cubes. We then select DINOv2 features from the source image and visualize the

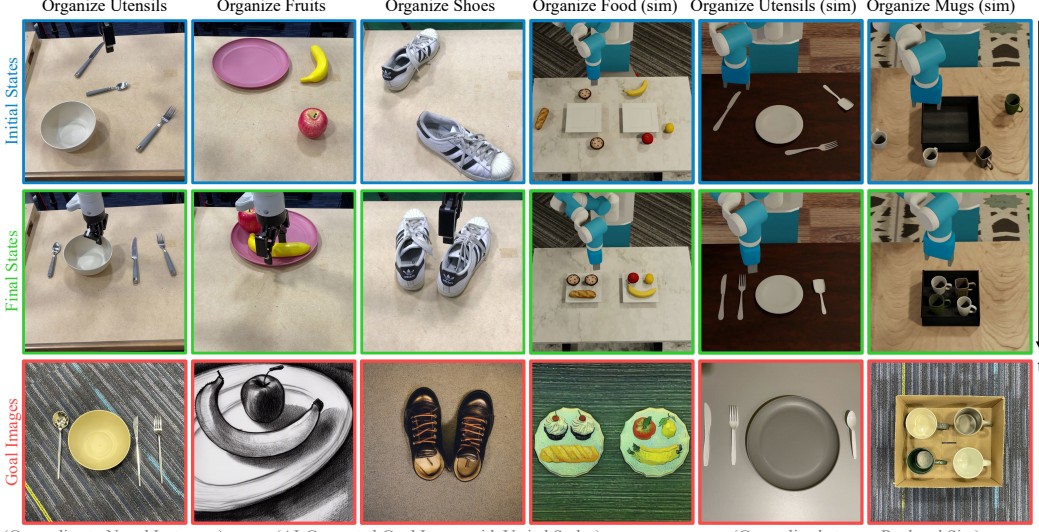

Figure 6: **Qualitative Results.** We qualitatively evaluate our proposed framework on rearrangement tasks, both in the real world and in simulation, encompassing tasks such as organizing utensils, fruits, shoes, food, and mugs. The figure highlights that our representation can generalize across varied instances, styles, and contexts. For instance, in the example of organizing fruit, the goal image, unlike the workspace, is styled as a sketch drawing. Because our representation encodes semantic features, the banana in the workspace can correspond with the banana in the goal image, which allows the task to be finished. This wide range of tasks showcases the generalization capabilities of our framework.

corresponding heatmap on the reconstructed feature mesh. We could see that our D³Fields could reconstruct the mesh with high quality and highlight semantically similar areas across different instances and contexts, while F3RM fails to reconstruct the mesh accurately, which leads to an inaccurate correspondence heatmap. Since FeatureNeRF is only pre-trained on a small dataset, it fails to construct an accurate mesh of the scene and find accurate correspondence in novel scenes, demonstrating our representation's effectiveness in reconstructing meshes and encoding semantic features. Quantitative comparisons are included in the supplementary material.

We also evaluate the time efficiency of constructing the implicit representation given four RGBD views across four scenes on the machine with A6000 GPU. Our method takes $0.166\pm0.002$ seconds, which is significantly more efficient than F3RM, which takes $88.379 \pm 5.306$ seconds.

### 4.3 Zero-Shot Generalizable Rearrangement

We conduct a qualitative evaluation of D³Fields in common robotic rearrangement tasks in a zero-shot manner, with partial results displayed in Figure 1 and Figure 6. We observed several key capabilities of our framework, which are as follows:

**Generalization to AI-Generated Goal Images.** In Figure 1, the goal image is rendered in a Van Gogh style, which is distinctly different from those in the workspace. Since D3Fields encode semantic information, capturing shoes with varied appearances under similar descriptors, our framework can manipulate shoes based on AI-generated goal images.

**Compositional Goal Images.** In the office desk organization task in Figure 1, the robot first arranges the mouse and pen to match the goal image. Then, the robot repositions the mug from the top of a box to the mug pad, guided by a separate goal image of the upright mug. This example illustrates that our system can accomplish tasks using compositional goal images.

**Generalization across Instances and Materials.** Figure 4 and Figure 1 also show our framework's ability to generalize across various instances and materials. For example, the debris collection in Figure 1 shows our framework's ability to handle granular objects. Figure 6 further shows our framework's instance-level generalization, where the goal instances distinct from ones in the workspace.

**Generalization across Simulation and Real World.** We evaluated our framework in the simulator, as shown in the utensil organization and mug organization examples in Figure 6. Given goal images from the real world, our framework can also manipulate objects to the goal configurations. This underscores our framework's generalization capabilities between simulation and the real world.

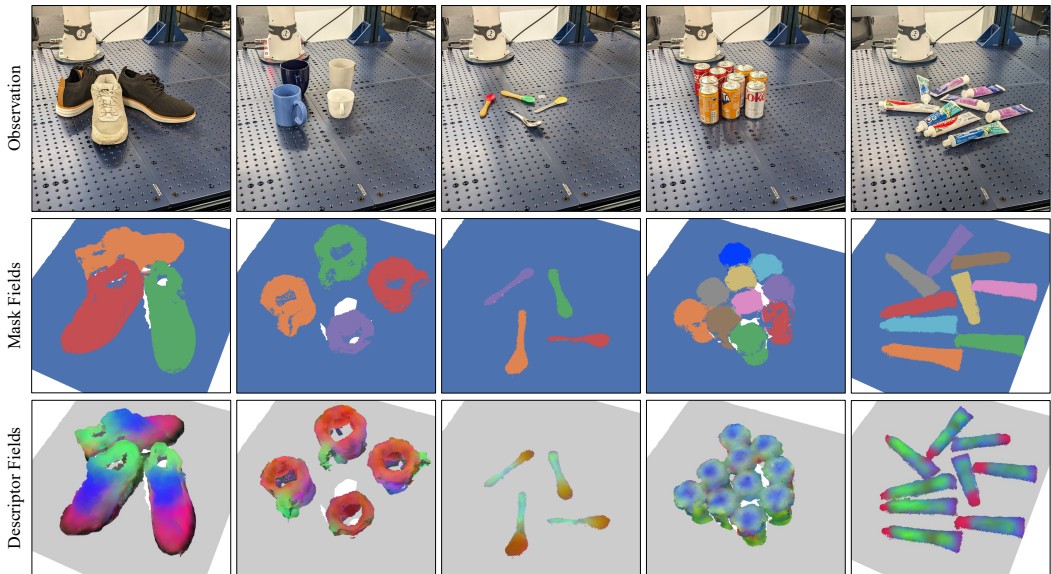

Figure 7: **Representation Visualizations.** To analyze our representation, we visualize the representation across different object categories. Mask fields color objects based on their instance masks, showing effective distinguishing among different instances. Descriptor fields color 3D points through PCA. The consistent color patterns across instances show that our representation could encode consistent semantic features over different instances. We also demonstrate our representation's robustness to clustered scenes in the right two columns.

### 4.4 D³Fields Analysis

To obtain a more thorough understanding of our D³Fields, we first extract the mesh using the marching cube algorithm. We evaluate vertices from the extracted mesh using our D³Fields and obtain the associated segmentation information and semantic features, as visualized in Figure 7. Mask fields in Figure 7 show a distinct 3D instance segmentation in different scenarios, even clustering scenes like cans and toothpaste. The 3D instance segmentation enables the downstream planning module to distinguish and manipulate multiple instances, as shown in the mug organization tasks.

Additionally, we visualize the semantic features by mapping them to RGB space using PCA. We observe that our semantic fields show consistent color patterns across different instances. In the provided shoe example, even though various shoes have distinct appearances and poses, they exhibit similar color patterns: shoe heels are represented in green, and shoe toes in red. We observed similar patterns for other object categories such as mugs and forks. The consistent semantic features across various instances help our manipulation framework to achieve category-level generalization. When encountering novel instances, our D³Fields can still establish the correspondence between the workspace and the goal image using semantic features. In addition, we analyze in the supplementary to show that D³Fields have better 3D consistency compared to simple point cloud stitching.

## 5 Conclusion

In this work, we introduce D³Fields, which implicitly encode 3D semantic features and 3D instance masks, and model the underlying dynamics. Our emphasis is on zero-shot generalizable rearrangement tasks specified by 2D goal images of varying styles, contexts, and instances. Our framework excels in executing a diverse array of robotic rearrangement tasks in both simulated and real-world scenarios. Its performance greatly surpasses baseline methods such as FeatureNeRF and F3RM in terms of efficiency and effectiveness.

**Limitation.**  Our method lifts 2D visual foundation models to the 3D world, enabling a range of zero-shot rearrangement manipulation tasks. However, we both benefit from and are limited by their capabilities. For example, the semantic feature field is not fine-grained enough to distinguish between the right and left sides of shoes. Visual foundation models with more fine-grained semantic features are needed. In addition, rearrangement tasks could fail when they need to follow a specific manipulation order to avoid collision with other objects, e.g. a crowded scene. In the future, incorporating a task planner to handle clustered scenes could be a valuable direction.

**Acknowledgments**

This work is partially supported by Sony Group Corporation, the Stanford Institute for Human-Centered AI (HAI), and Google. The opinions and conclusions expressed in this article are solely those of the authors and do not reflect those of the sponsoring agencies.

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

# D³Fields: Dynamic 3D Descriptor Fields for Zero-Shot Generalizable Rearrangement

Yixuan Wang[1*], Mingtong Zhang[2*], Zhuoran Li[3*], Tarik Kelestemur[4],
Katherine Driggs-Campbell[2], Jiajun Wu[5], Li Fei-Fei[5], Yunzhu Li[1]

[1]Columbia University, [2]University of Illinois, Urbana-Champaign, [3]National University of Singapore,
[4]Boston Dynamics AI Institute, [5]Stanford University

## Contents

# 1 Detailed Comparisons with Prior Works

## 1.1 Comparisons with Prior 3D Representations using Visual Foundational Models

We extend the application of state-of-the-art visual foundational models to robotic manipulation tasks, offering the following improvements over prior methods:

- **Efficiency**: Several existing works construct 3D representations from 2D models, but they are often time-consuming due to the distillation process. This limits their applicability in tasks requiring frequent visual feedback. For instance, prior works such as F3RM [1] and LERF [2] require 1.3 to 8 minutes for training. In contrast, our method takes only 0.166 seconds, making it 500 to 3,000 times faster.

- **Sparse Views**: Some existing works, such as F3RM and LERF, require dense camera views (more than 50) to build a high-quality 3D representation, which is not suitable for typical robotic manipulation workspaces with sparse camera views (1-4). In contrast, we have developed a more efficient and effective pipeline that uses only four views.

- **Planning Using Reconstructed Fields**: We explicitly use the reconstructed fields to find matches that define our planning objective, allowing us to employ model-based planning to solve rearrangement tasks. This contrasts with existing methods like F3RM, which require expert demonstrations for every new task.

Additionally, our work distinguishes itself from FeatureNeRF by applying our representation specifically to robotic manipulation tasks, whereas FeatureNeRF does not focus on manipulation [3].

## 1.2 Comparisons with Dense Object Nets

Our approach differs from Dense Object Nets (DON) in two major ways [4]:

- **Training Requirements**: DON is trained via contrastive learning, which requires foreground and background separation and a careful selection of positive and negative pairs. This means that deploying a DON in a new environment or on new object categories requires additional training efforts. In contrast, our pipeline does not need extra training and can be applied to a diverse set of object categories by leveraging visual foundational models.

- **Generalization**: DON is trained on small-scale datasets and may not generalize well to new object categories and scenarios. Our pipeline, however, offers stronger generalization capabilities without the need for retraining and additional data collection.

# 2 Method Details

## 2.1 Notation: Camera Transformation and Projection

We assume that there are multiple RGBD cameras with fixed viewpoints. We assume all cameras' intrinsic parameters $\mathbf{K}$ and extrinsic parameters $\mathbf{T}$ are known. The $i$th camera's extrinsic parameters are defined as follows:

$$\mathbf{T}_i = \begin{bmatrix} \mathbf{R}_i & \mathbf{t}_i \\ 0^T & 1 \end{bmatrix} \in \mathbb{SE}(3), \tag{1}$$

where Euclidean group $\mathbb{SE}(3) := \{\mathbf{R}, \mathbf{t} \mid \mathbf{R} \in \mathbb{SO}^3, \mathbf{t} \in \mathbb{R}^3\}$. For a 3D point $\mathbf{x}$ in the world frame, we could obtain the pixel $\mathbf{u}_i$ projected in $i$th camera and distance to $i$th camera $r_i$ as follows:

$$\mathbf{u}_i = \text{Proj}\left(\mathbf{K}_i \left(\mathbf{R}_i \mathbf{x} + \mathbf{t}_i\right)\right), \quad r_i = [0, 0, 1]^T \left(\mathbf{R}_i \mathbf{x} + \mathbf{t}_i\right), \tag{2}$$

where Proj performs perspective projection, mapping a 3D vector $p = [x, y, z]^T$ to a 2D vector $q = [x/z, y/z]^T$.

---

**Algorithm 1** Fusion Process

---

1: **procedure** FUSION(**x**)                                                                            ▷ Input 3D point
2:     $\mathbf{u}_i, r_i \leftarrow \text{Project}(\mathbf{x}, i), r'_i \leftarrow \mathcal{R}_i[\mathbf{u}_i]$                           ▷ 3D projection and depth reading
3:     $d_i \leftarrow r'_i - r_i, d'_i \leftarrow \text{Truncate}(d_i, \mu)$ ▷ Truncated depth difference using Eq. 3 in main paper
4:     $v_i, w_i \leftarrow \text{Weights}(d_i)$                             ▷ Assign weights to each view using Eq. 4 in main paper
5:     $\mathbf{f}_i, \mathbf{p}_i \leftarrow \text{Interpolate}(\mathcal{W}_i^{\mathbf{f}}, \mathcal{W}_i^{\mathbf{P}}, \mathbf{u}_i)$           ▷ Interpolate features using Eq. 5 in main paper
6:     $\mathbf{f}, \mathbf{p} \leftarrow \text{Fuse}(\mathbf{f}_i, \mathbf{p}_i, v_i, w_i)$                             ▷ Fuse features using Eq. 6 in main paper

---

## 2.2 Fusion Equation Explanation

Here is a detailed explanation for Equation 6.

- Signed Distance $d$: Assuming $\sum v_i \neq 0$ and ignoring $\delta$, the equation simplifies to $d = \frac{\sum v_i d'_i}{\sum v_i}$. In this case, the weights sum to 1.

- Semantic Feature $\mathbf{f}$ and Instance Masks $\mathbf{p}$: Similarly, with $\sum v_i \neq 0$ and ignoring $\delta$, we have $\mathbf{f} = \frac{\sum v_i w_i f'_i}{\sum v_i}$ and $\mathbf{p} = \frac{\sum v_i w_i p'_i}{\sum v_i}$. The weight $w_i$ varies based on proximity to the surface: $w_i = 1$ when $\mathbf{x}$ is close, summing weights to 1, and $w_i$ approaches 0 when $\mathbf{x}$ is far, causing $\mathbf{f}$ and $\mathbf{p}$ to converge to 0.

## 2.3 Correspondence Equation Explanation

Equation 8 describes the process of finding the correspondence from $j$th sampled point's associated feature $\mathbf{f}_j^0 \in \mathbb{R}^f$ to a 2D point $\mathbf{s}_{\text{goal},j}$ in the image space. The computation process consists of three steps:

- $\alpha_{ij}$ is the feature-space distance between $j$th sampled point and $i$th goal image pixel. We first extract goal image's feature map $\mathcal{W}_{\text{goal}}$ and read out $i$th pixel's corresponding feature $\mathcal{W}_{\text{goal}}[\mathbf{u}_i]$. Then we compute the L2 distance between $\mathcal{W}_{\text{goal}}[\mathbf{u}_i]$ and $\mathbf{f}_j^0$ and assign to $\alpha_{ij}$.

- After applying the softmax to $\alpha_{ij}$ over the whole image, we obtain $\beta_{ij}$. The summation of $\beta_{ij}$ over the image space, i.e. $\sum_{i=1}^{H \times W} \beta_{ij} = 1$.

- Then we apply the weighted sum of pixels to obtain the corresponding pixel on the goal image $\mathbf{s}_{\text{goal},j}$.

## 2.4 Grounded-SAM Masks Association

Using Grounded-SAM [5, 6], we could extract instance segmentation masks from each view. However, masks from different views can contain a different number of instances, and the instance IDs may not be consistent. To tackle this problem, we need to post-process the instance segmentation results. The high-level idea is to merge instances from different viewpoints based on their geometric distance.

We will save all merged instances to a list. Specifically, we will first start from the first viewpoint. For each instance mask, we map them to 3D point clouds and save them into a list. Then, we move to the next viewpoint and map all instance masks to 3D point clouds. We will compare each instance with the merged instances in the list. If they have significant overlap, measured by the Intersection of Union (IoU) of the two point clouds, we will merge the instance from the new viewpoint with the merged instances in the list. This process will continue until all viewpoints have been iterated through.

After merging all instances, we will filter out instances that are not stably detected. Specifically, instances that meet one of the following criteria will be filtered out:

- The instance has little point cloud.
- The instance is known to be a background, such as the table.

- The instance overlaps with other instances, while other instances have a higher confidence.

After filtering, we will assign consistent instance IDs to the instance masks in each viewpoint.

## 2.5 Mask Tracking using XMem

After using Grounded-SAM to obtain initial masks, we use XMem to track instance masks in later frames so that we can skip time-consuming segmentation detection for subsequent frames. By applying XMem, we significantly reduce the computational cost and processing time, making the pipeline more efficient.

## 2.6 Dynamics Training Details

We instantiate the dynamics model $f(\cdot, \cdot)$ as graph neural networks (GNNs) that predict the evolution of particles $\boldsymbol{s}^t \in \mathbb{R}^{2 \times n_s}$ under external actions $a_t$. We also construct edges $\boldsymbol{e}^t \in \mathbb{N}^{2 \times n_e}$ according to particle distance, where $\boldsymbol{e}_j^t = (u_j^t, v_j^t)$ represents an edge connecting from particle $u_j^t$ to particle $v_j^t$. $f(\cdot, \cdot)$ consists of node and edge encoders $f_{\mathcal{O}}^{\text{enc}}(\cdot, \cdot)$, $f_{\mathcal{E}}^{\text{enc}}(\cdot, \cdot)$ to obtain node and edge representations:

$$
\begin{aligned}
p_i^t &= f_{\mathcal{O}}^{\text{enc}}(\boldsymbol{s}_i^t, a_t), \quad i = 1, \ldots, n_{\boldsymbol{s}}, \\
q_j^t &= f_{\mathcal{E}}^{\text{enc}}(\boldsymbol{s}_{u_j^t}^t, \boldsymbol{s}_{v_j^t}^t), \quad j = 1, \ldots, n_{\boldsymbol{e}}.
\end{aligned}
\tag{3}
$$

Then, we use node and edge decoders $f_{\mathcal{O}}^{\text{dec}}(\cdot, \cdot)$, $f_{\mathcal{E}}^{\text{dec}}(\cdot, \cdot)$ to predict the next time step's particle states:

$$
\begin{aligned}
r_j^t &= f_{\mathcal{E}}^{\text{dec}}(q_j^t), \quad j = 1, \ldots, n_{\boldsymbol{e}}, \\
\hat{\boldsymbol{s}}_i^{t+1} &= f_{\mathcal{O}}^{\text{dec}}(p_i^t, \sum_{j \in \mathcal{N}_i} r_j^t), \quad i = 1, \ldots, n_{\boldsymbol{s}},
\end{aligned}
\tag{4}
$$

where $\mathcal{N}_i$ is the index set of the edges that connect to particle $i$. In practice, we follow Li et al. [7] and use multi-step message passing over the graph to approximate the propagation of action impacts.

## 2.7 Keypoints Tracking Initialization

To initialize keypoint tracking, we first densely sample points, which can either come from grid sampling or instances' 3D point clouds. Then, we evaluate these points using our D$^3$Fields and mask out those not belonging to the desired instance. Finally, we downsample these points to the desired number using farthest point sampling.

## 2.8 Model-Predictive Control (MPC) Details

As described in Section 3.4 of the main paper, our MPC framework needs a reference camera to bridge the gap between 3D representation and 2D representation. In our work, the reference camera's extrinsic parameters are manually defined according to the tasks. Typically, it looks down at the workspace from above. Its intrinsic parameters are the same as the real camera's intrinsic parameters. The detailed MPC algorithm we used is described in Algorithm 2.

For the pick-and-place tasks, our dynamics model is simplified, as the object is rigidly attached to the end-effector.

## 2.9 Discussion of Backbone Choice

We choose DINOv2 as our feature backbone because there are several great and unique properties of DINOv2, including:

- Generalization: DINOv2 has demonstrated consistent feature extraction across diverse object categories and scenes, which allows us to apply our pipeline to various object categories and bridge the gap between goal image and workspace.

---
**Algorithm 2** Trajectory optimization at each MPC step
---
**Input:** Current state $\mathbf{s}_0$, goal $\mathbf{s}_{\text{goal}}$, time horizon $T$, gradient descent iteration $N$
    the perception module $h$, and the dynamics module $f$
**Output:** Actions $a_{0:T-1}$

Sample current action sequence $\hat{a}^*_{0:T-1}$
**for** $i = 1, \ldots, N$ **do**
    Sample $M$ action sequences $\hat{a}^{1:M}_{0:T-1}$ near current action sequence
    **for** $m = 1, \ldots, M$ **do**
        **for** $t = 0, \ldots, T - 1$ **do**
            Predict the next step $\mathbf{s}_{t+1} \leftarrow f(\mathbf{s}_t, \hat{a}^m_t)$
        Calculate the task loss $c^m \leftarrow c(\mathbf{s}_T, y_g)$
    Calculate the current action sequence $\hat{a}^*_{0:T-1}$ using the task loss $c^{1:M}$
Return $\hat{a}^*_{0:T-1}$
---

- Refined Correspondence: DINOv2 demonstrates the capability to establish a refined correspondence, even when two images have quite different backgrounds and contexts. This is important for our tasks since we need refined correspondence between the goal image and the workspace to define the objective function for the manipulation.

However, our framework is not specific to DINOv2. If there is a better visual foundational model in the future, we could replace it easily in a plug-and-play manner.

## 3 Additional Experiments

### 3.1 Implementation Details

For the truncation threshold $\mu$, we set it to 0.02 across all experiments. For the prompts used for Grounded-SAM [5, 6] in Figure 7 in the main paper, from left to right, they are "shoe", "mug", "spoon", "can", and "toothpaste" respectively. The confidence threshold used for Grounded-SAM is 0.2.

We also compare with several baselines, with details listed below:

- Dense Object Nets (DON) [4]: We compare the effects of using different feature backbones, with DON as one of the baseline backbones. We use the pre-trained DON model since our method also uses off-the-shelf models with no re-training or additional data. Specifically, we use the model trained on the shoe class, which can be found here.

- DINO [8]: Another feature backbone we baseline on is DINO, which is the precursor to DINOv2. We use the code provided by [9] to extract dense DINO features.

- RGBD+DINOv2: We also compare our method to simply merging point clouds with DINOv2 features from multiple viewpoints [10].

- FeatureNeRF [3]: We compare our representation with other state-of-the-art 3D implicit semantic representations, including FeatureNeRF. We trained the model on the car example dataset provided by the authors. For comparison with our model, we do not distill the model from the DINO model but from DINOv2. It is worth noting that FeatureNeRF only uses one RGB image to generate the neural fields. We found that providing more views to the FeatureNeRF model during inference time leads to worse performance. Therefore, we keep its input as a single-view RGB image.

- Distilled Feature Fields (F3RM) [1]: Another 3D implicit semantic representation we compare to is F3RM. We use the same training code provided by the authors, except that we distill from DINOv2 models to make it comparable with our model. In addition, we use four camera views as F3RM inputs instead of dense views as the original paper.

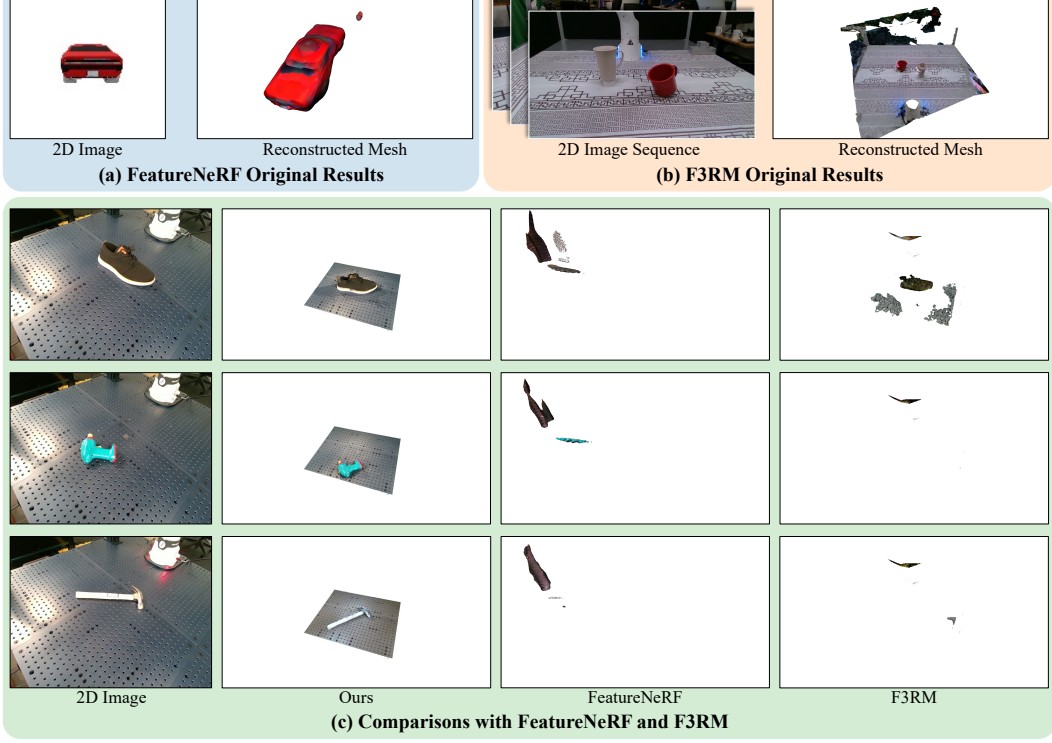

Figure 1: **Mesh Reconstruction Comparison.** (a) shows the reconstructed mesh of the FeatureNeRF, given a 2D image from the training distribution. This reflects that our mesh extraction process works well when the input image is within the training distribution. Given a sequence of 2D images densely scanning the workspace, (b) also shows good reconstruction quality of the scene. However, when given sparse views containing novel instances, both FeatureNeRF and F3RM fail to generate accurate meshes for the scene, which demonstrates the effectiveness of our method.

For the dynamics training of the shoe-pushing example, we collected 20 episodes of pushing one shoe. Then, we trained a dynamics model that can take in current particles and a pushing action and predict particles in the next step.

For the evaluation in the real world, we summarize the details of our tasks in Table 1.

| Environment | Task Name | Objects |
|---|---|---|
| Real World | Organize Shoes | Shoe |
| | Collect Debris | Almonds |
| | Organize Office Table | Mouse, Pen, Mug |
| | Organize Utensils | Knife, Spoon, Fork, Bowl |
| | Organize Fruits | Apple, Banana |
| | Push Shoes | Shoe |
| Simulation | Serve Food | Cupcake, Bread, Tomato, Lemon, Banana |
| | Organize Mugs | Mug |
| | Organize Shoes | Shoe |
| | Organize Utensils | Knife, Spoon, Fork |

Table 1: **Task Details Summary.** This table summarizes our task environment, specific tasks, and objects. We evaluate our framework on eight tasks and fifteen object categories, where each object category covers several object instances with diverse appearances and shapes.

## 3.2 Keypoint Tracking Results

We show two examples of 3D keypoint tracking in Figure 2. In the first scenario, we track a shoe as it is pushed and subsequently flipped. The second example demonstrates tracking a shoe that is lifted and then placed down. Our system robustly tracks the shoe in 3D space. These examples underscore the effectiveness of D³Fields in maintaining accurate tracking in dynamic scenarios, which enables our dynamics learning capabilities.

## 3.3 Mesh Comparisons with FeatureNeRF and F3RM

We qualitatively compare the descriptor fields generated by the three methods. We extract the mesh from these fields using marching cubes, as shown in Figure 1. We observe that our D³Fields could generate accurate color meshes given sparse views. FeatureNeRF could reconstruct a reasonable mesh given a single 2D image from the training distribution, as shown in Figure 1 (a). However, when it encounters a new object outside the training distribution, the reconstructed mesh will be completely off, even when we apply the image preprocessing to align the testing images with the training set in terms of image sizes, data range, and background color. Although Figure 1 (b) shows that we can reconstruct a clear mesh with dense image sequences as in the original paper, its color mesh is quite inaccurate given sparse viewpoints in our experiment setting, except for the shoe case.

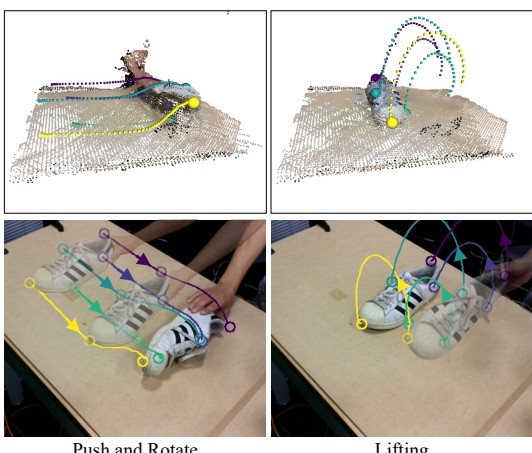

Push and Rotate          Lifting

Figure 2: **Keypoint Tracking.** We apply D³Fields to tracking tasks and showcase two tracking examples, both of which involve 3D motions and partial observations from single viewpoints. This shows that our representation is 3D, dynamic, and semantic.

## 3.4 Quantitative Correspondence Comparisons with FeatureNeRF and F3RM

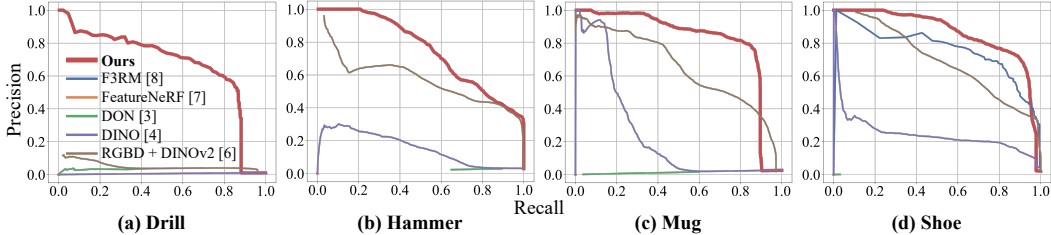

Figure 3: **Precision-Recall of Various Thresholds for Different Instances.** The curves show how D³Fields compares with 3 baseline methods in terms of matching quality, tested on 4 different instances: mug, bag, pan, and shoe. We use the precision-recall curve to measure the correspondence quality. Our method shows to consistently exceeds the performance of the baseline approaches, which demonstrates our method's capability to encode semantic information accurately and establish precise correspondences using the semantic information.

In addition, we also measure the quantitative correspondence accuracy of our method, FeatureNeRF, and F3RM. We manually label the ground truth correspondence keypoints on the source image and the target descriptor fields. We measure the correspondence quality using the precision-recall curve, as shown in Figure 3 A larger area under the curve indicates better correspondence quality. Details regarding the precision-recall curve are provided later. We could see that F3RM and FeatureNeRF collapse to the origin point except for the shoe example for F3RM. This is because these two methods fail to reconstruct meshes given sparse observations and unseen instances. In contrast, our method shows a much better correspondence quality.

To generate the precision-recall curve, we manually label one point $\mathbf{x}_{\text{src}}$ on the 2D source image, and a set of corresponding 3D points $\mathbf{x}_{\text{tgt}}$. For 2D points, we obtain the associated semantic feature $\mathbf{f}_{\text{src}}$. For vertices on the reconstructed mesh, we can obtain a set of semantic features $\{\mathbf{f}_{\text{tgt},0}, ..., \mathbf{f}_{\text{tgt},N}\}$. Then we compute the cosine similarity between features $\mathbf{f}_{\text{src}}$ and $\{\mathbf{f}_{\text{tgt},0}, ..., \mathbf{f}_{\text{tgt},N}\}$. For one similarity threshold $\tau$, we filter out a set of points $\mathbf{F}_\tau$ with similarity scores higher than $\tau$. Additionally, we identify the set of points $\mathbf{G}$ that are close to $\mathbf{x}_{\text{tgt}}$. We then define precision $P_\tau$ and $R_\tau$ as follows:

$$P_\tau = \frac{|\mathbf{F}_\tau \cap \mathbf{G}|}{|\mathbf{F}_\tau|}, \quad R_\tau = \frac{|\mathbf{F}_\tau \cap \mathbf{G}|}{|\mathbf{G}|}. \tag{5}$$

By varying $\tau$, we can plot the precision-recall curve as shown in Figure 3.

### 3.5 Comparisons with F3RM under Dense Views

We compare our approach using 4 views against F3RM [1], which uses 50 views (with and without Grounding DINO + SAM supervision), on the Dense Object Nets dataset [4].

Our qualitative results are shown in Figure 4. Similar to the main paper, We compare the reconstructed mesh, descriptor fields, and mask fields for our approach (4 views), F3RM without Grounding DINO + SAM (50 views), and F3RM with Grounding DINO + SAM (50 views). We observed that F3RM with 50 views reconstructs high-quality meshes, confirming our F3RM implementation is bug-free. Additionally, there is no significant qualitative difference between our method with 4 views and F3RM with 50 views, demonstrating the effectiveness of our representation. Lastly, we did not observe notable differences between F3RM with and without Grounding DINO + SAM, which is consistent with our previous conclusion that this additional supervision does not significantly contribute to F3RM training.

Similar to our paper, we also evaluated quantitative correspondence quality using the Precision-Recall (P-R) curve, as shown in Figure 5. A larger area under the P-R curve indicates better correspondence quality. Our results show no significant correspondence differences between our approach with 4 views and F3RM with 50 views, indicating that our representation is effective even with sparse views.

### 3.6 Ablation Study: Qualitative Correspondence Comparisons

In this section, we first study how different feature backbones could affect the correspondence quality. Then we show the qualitative correspondence results of our method. As mentioned in Section 3.1, we substitute our method's backbone with other pre-trained models, like DON and DINO [4, 8]. We also compare with RGBD+DINOv2 to demonstrate its effectiveness.

Figure 6 shows the qualitative correspondence results of all ablation baselines. There are three key observations we can make from this figure.

- Compared with RGBD+DINOv2, our method's correspondence quality is better. We achieve more accurate correspondence since our representation can amortize noise from single views by considering 3D consistency. Although RGBD+DINOv2 can achieve some spatial consistency, there are still variances in results from different viewpoints, while our method guarantees spatial consistency.

- Compared with DINO, our correspondence is more fine-grained and accurate. Thanks to the advancements in foundational visual models, DINOv2 encodes more fine-grained features and enables correspondence with higher accuracy.

- DON struggles to generalize to novel scenes and unseen object categories. Although the original DON shows good correspondence quality, it is trained on one type of object with a relatively small dataset. Compared with visual foundational models, it shows limited generalization capabilities in terms of scenes and object categories.

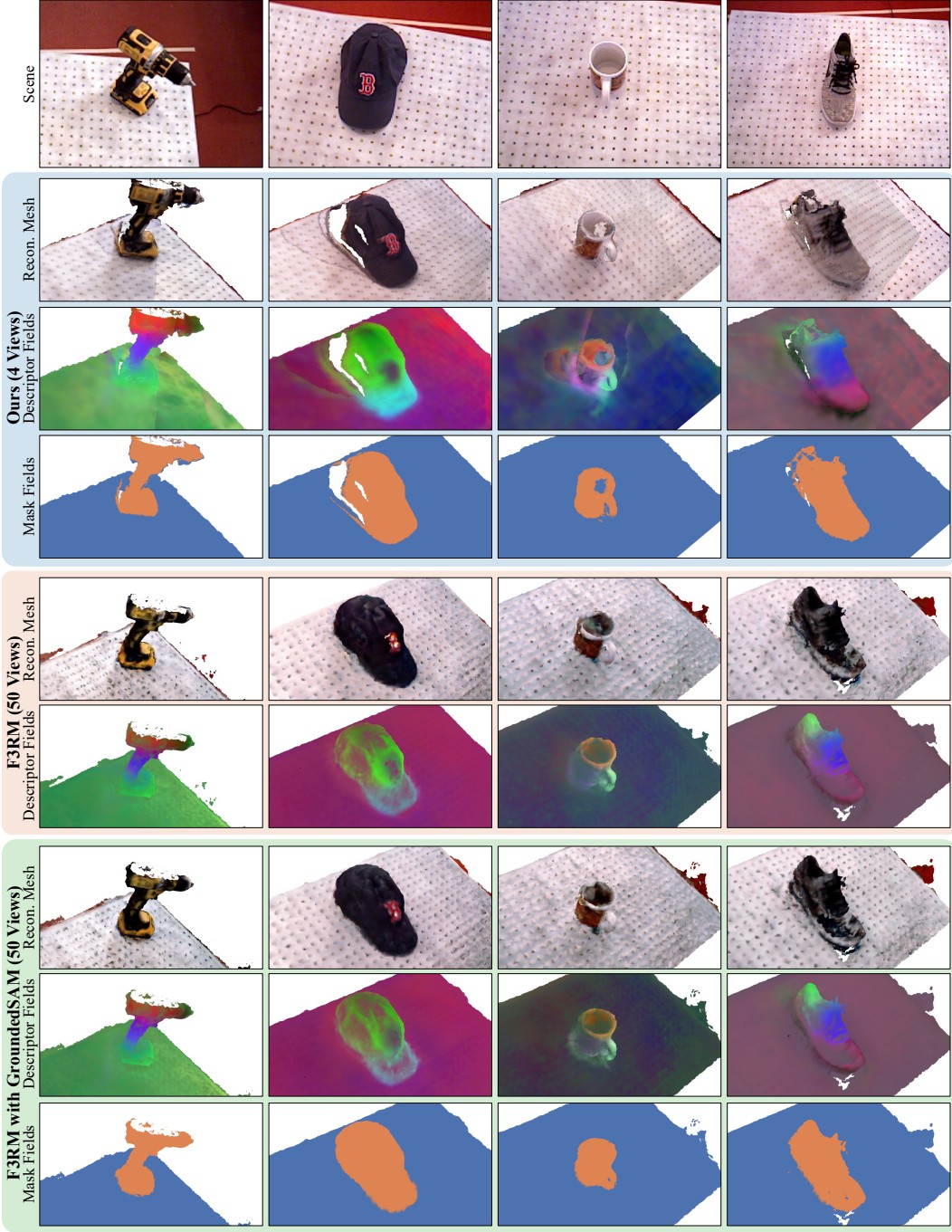

Figure 4: **Comparisons with F3RM under Dense Views.** We compared the reconstructed mesh, feature fields, and mask fields with F3RM using dense views. Our results showed that, despite using sparse inputs, our method did not exhibit significant qualitative performance degradation compared to F3RM under dense views, demonstrating the efficiency of our approach.

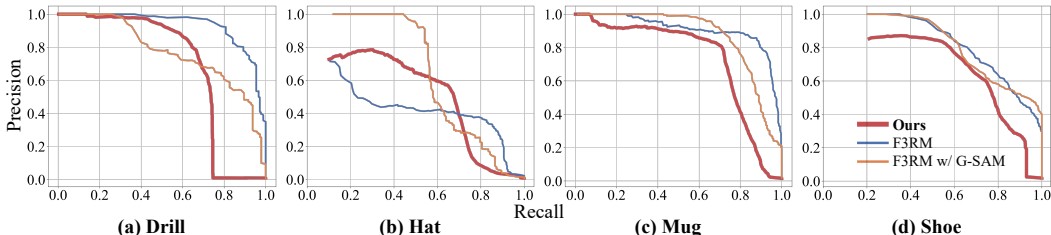

**Figure 5: Quantitative Comparisons with F3RM under Dense Views.** We use the P-R curve to measure the correspondence accuracy. Our results show no significant differences in correspondence between our approach with 4 views and F3RM with 50 views, indicating that our representation remains effective even with sparse views.

We also visualize the correspondence from 2D images to our workspace as shown in Figure 7. Specifically, we extract the DINOv2 feature of the selected pixel in the 2D image. Then we highlight the part of the 3D mesh with features close to the query feature. There are two observations regarding the qualitative correspondence results. First, the semantically similar parts are correctly matched across different instances and contexts. For example, when we select the rim of the plate in the 2D image, the corresponding part in the 3D mesh is highlighted. This matching is consistent across different object parts, such as the head and tail of the shoe, the handle and blade of the knife, and the tip and bar of the drill. Second, the correspondence is multimodal when there are multiple semantically similar object parts in the workspace. For example, when we select the spoon handle in the 2D image, multiple utensil handles in the workspace are highlighted. The correspondence qualitative results show that our $D^3$Fields could establish meaningful correspondences across different instances and contexts, so that we can rely on correspondence to define the planning objective function.

### 3.7 Ablation Study: Quantitative Correspondence Comparisons

Similar to the main paper, we generate the precision-recall curve to quantitatively compare the correspondence quality with ablation baselines. We can make the following observations regarding the baseline correspondence results.

- Compared with RGBD+DINOv2, our method shows more accurate correspondence results. This is because our $D^3$Fields can average out noise from each viewpoint, while RGBD+DINOv2 accumulates noises.
- DINO faces challenges in accurately distinguishing specific object components. This limitation results in less precise correspondence, as shown in Figure 3.
- Although DON can encode semantic features in seen environments and instances, it fails to generalize to novel environments and object categories. Therefore, its correspondence results are even worse than DINO, as shown in Figure 3.

### 3.8 Abaltion Study: Quantitative Manipulation Results

In Figure 8 (a), we measure performance using the IoU between the mask of the goal image and the mask of the final state post-manipulation. Higher IoU values indicate a greater degree of alignment between the intended and achieved configurations. Our method demonstrates superior performance across five distinct object categories, consistently outshining the baseline methods. For each category, we performed 5 experiments for the evaluation results. This not only highlights its exceptional manipulation accuracy but also its robust generalization capabilities. While the DINO model exhibits some struggles, particularly in distinguishing specific object components and consequently yielding less precise results, it still performs better than DON. Although DON shows commendable results with familiar objects and configurations, its performance dips in novel scenarios, revealing a lack of generalization. These results collectively emphasize the significant advantages of our method in diverse and accurate object manipulation.

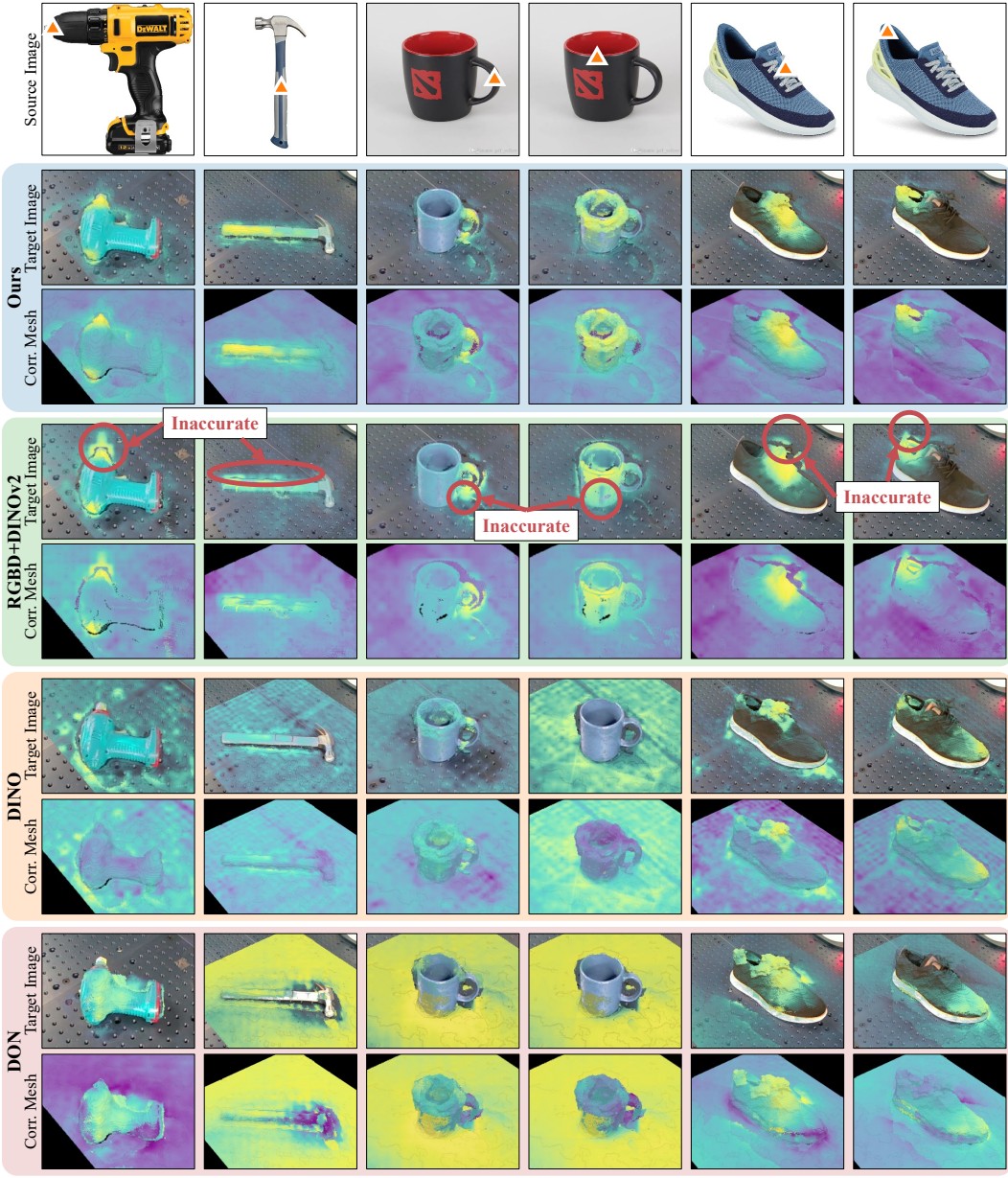

Figure 6: **Correspondence Quantitative Comparison.** The top row shows the selected pixel from the source image, and the following rows show the corresponding areas for different methods. While our method could have accurate correspondence, RGBD+DINOv2 corresponds to some points in the background. For example, the drill tip is not accurately highlighted in the RGBD+DINOv2 example, while ours can accurately highlight the drill tip. Ours with DINO feature backbones fail to identify objects accurately, while ours with DON fail to generalize to novel scenes and novel instances.

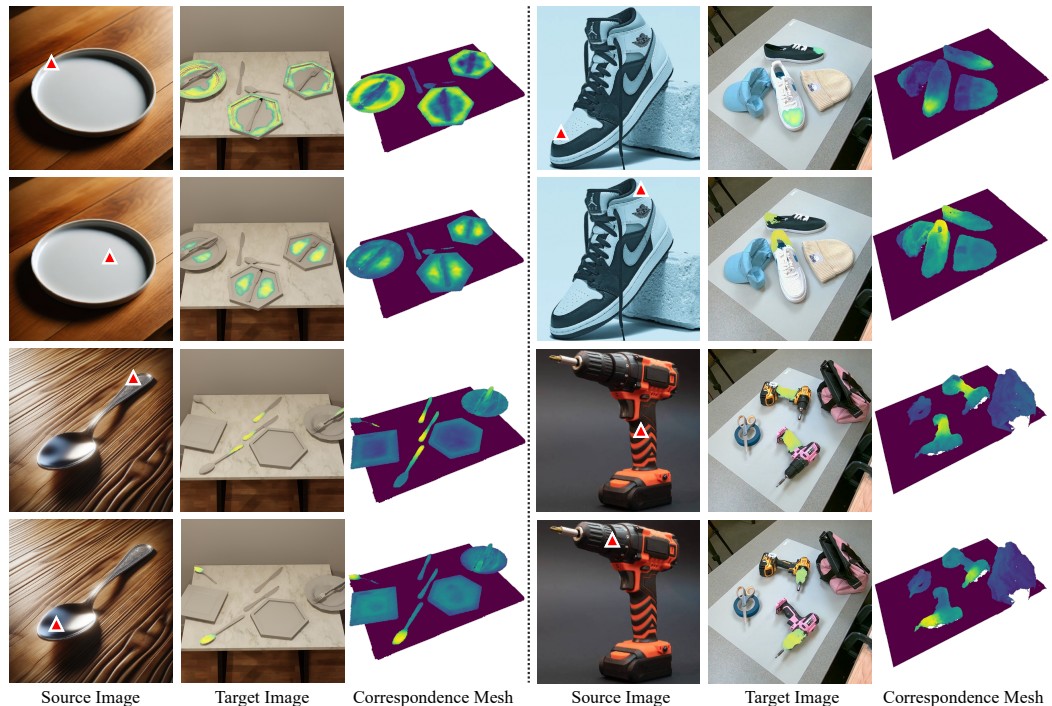

| Source Image | Target Image | Correspondence Mesh | Source Image | Target Image | Correspondence Mesh |

Figure 7: **Cross-Domain Correspondence.** The red triangles represent query points in the source image, and the corresponding areas are highlighted in the 3D mesh. First, we observe that our representation can encode features for object parts and establish the correspondence, such as spoon tips and spoon handles. In addition, we found the correspondence can be multimodal. When the shoe head is selected, multiple shoe heads in the workspace are highlighted. At last, the correspondence is generalizable across different contexts, instances, and domains, which demonstrates our method's generalization capabilities.

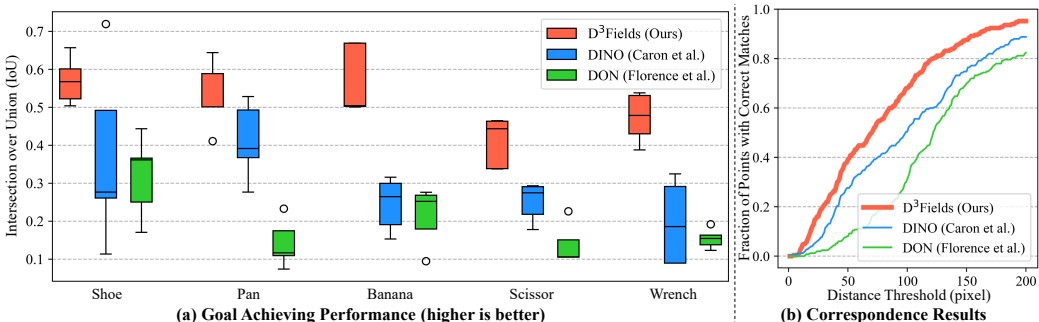

Figure 8: **Quantitative Evaluation.** We perform real-world quantitative evaluations by measuring final goal-achieving performance and keypoints correspondence accuracy. (a) We use IoU to measure goal-achieving performance. Results indicate that our method aligns with the goal configurations much better than DON and DINO across various object categories and scenarios. (b) We measure the keypoints correspondence accuracy according to the fraction of points with accurate matches, with correct matches determined by a distance threshold. Our method is consistently better at aligning with the goal image, regardless of the chosen threshold.

In Figure 8 (b), we present the correspondence results. We label 10 corresponding keypoint pairs on both the goal image and the final manipulation result to sufficiently evaluate the correspondence accuracy. The accuracy of correspondence was determined by calculating the proportion of keypoints that were accurately matched, using a predefined distance threshold as the criterion. If the distance between corresponding keypoints exceeds this threshold, they are determined as unmatched. Our method shows superior performance across various thresholds, consistently outperforming the baseline models. DINO emerges as the second-best in terms of performance, exhibiting broad applicability but with a lower precision compared to our method. Meanwhile, DON lags in performance, primarily due to its struggles with generalization in novel scenarios. These results, in conjunction with those from Figure 8 (a), reiterate our method's outstanding capabilities in both generalization

and accuracy. While DINO provides reasonable applicability, it lacks the precision of our approach, and the performance of DON is hindered by its limited adaptability.

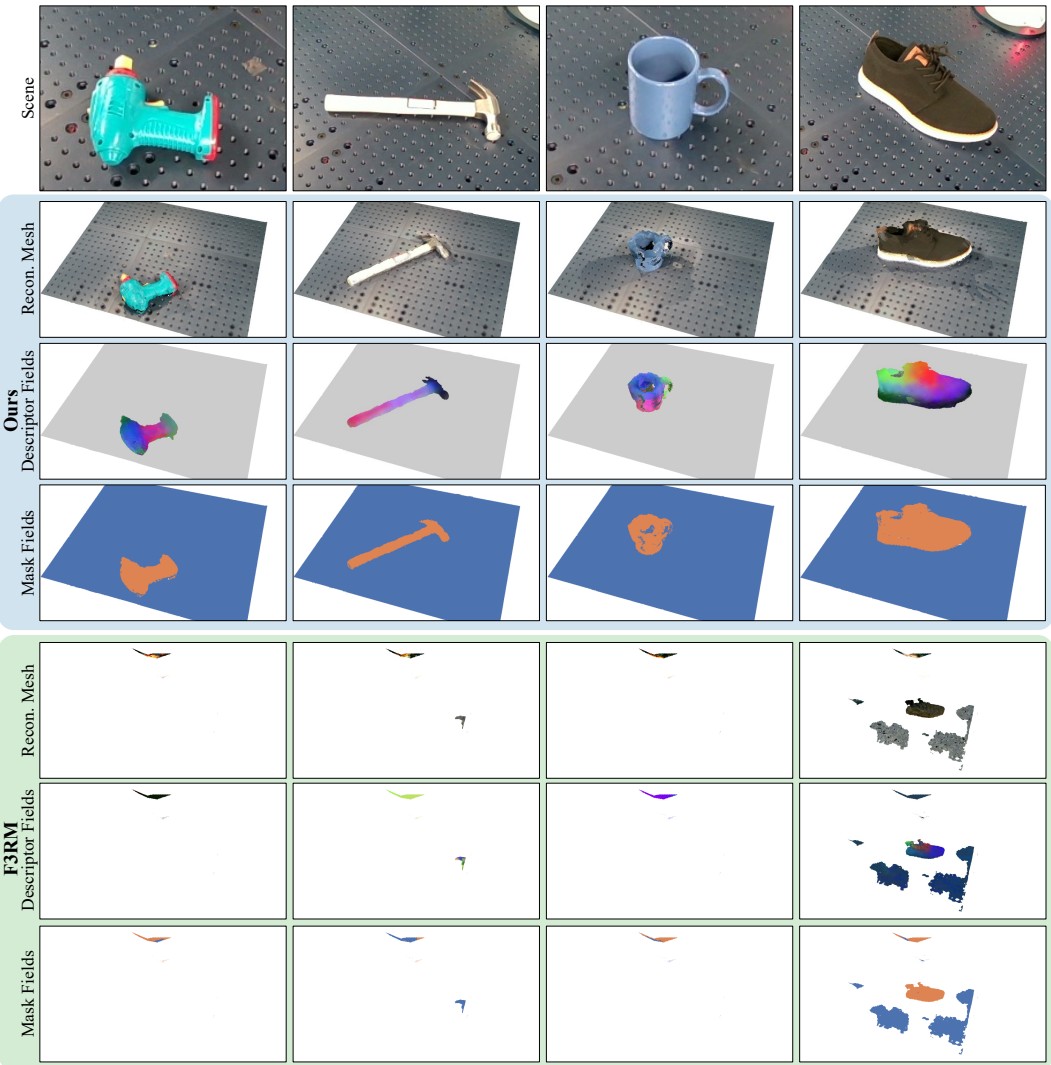

Figure 9: **Qualitative Comparison with F3RM.** We compare our representation with F3RM trained with additional Grounded-SAM on four scenes. We qualitatively evaluate their performance by reconstructing their mesh and visualizing descriptor fields and mask fields. Although Grounded-SAM supervision can help F3RM to segment out objects as shown in the shoe example, it does not contribute too much to reconstructing a quality mesh. In contrast, our representation can consistently reconstruct meshes and generate quality descriptor fields and mask fields across four scenes.

### 3.9 F3RM with Grounded-SAM

We also trained F3RM [1] on Grounded-SAM [5, 6] and compare with our method, as shown in Figure 9. For F3RM, we use the same setting as the main paper, except for the additional Grounded-SAM labeling. We reconstruct meshes and visualize the corresponding descriptor meshes and mask meshes, similar to what we did in the main paper. We have the following observations:

- F3RM could segment out objects given additional Grounded-SAM supervision, as shown in the shoe example. This indicates that our F3RM training pipeline works as expected.

- Additional Grounded-SAM supervision does not influence too much on results. We could see that reconstructed meshes are quite similar to the results from the main paper because

additional Grounded-SAM supervision does not contribute too much to NeRF training. Instead, sparse camera views are the main reason for the poor performance of F3RM.

- Our representation can consistently construct quality meshes and generate clear descriptor fields and mask fields, which demonstrates the effectiveness of our method.

### 3.10 Debris Experiment Details

In this section, we provide more details and visualization for our debris experiments. Our initial multi-view observations and goal images are visualized in Figure 10. In this task, we want the robot to push spreading almonds into one object pile. We could see that initially, the debris spreads over the workspace, which is quite different from the goal states. This task is challenging since the dynamics of object piles is hard to predict. In addition, such manipulation capability needs an efficient perception module so that it can gain visual feedback from the environment. Our zero-shot rearrangement framework can collect these almonds into one pile successfully using our dynamic representation and the learned dynamics model.

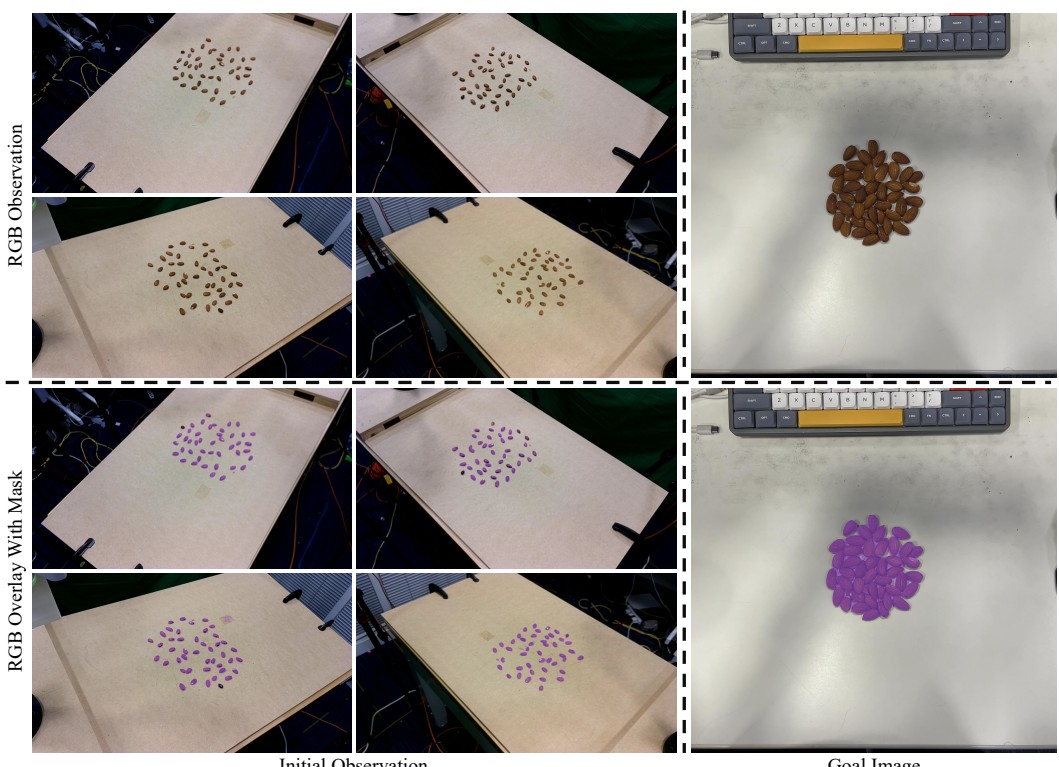

Figure 10: **Debris Experiments Details.** We visualize initial multi-view observations in our debris experiments and the corresponding goal image. The bottom row overlays the RGB observation with the mask. In this task, our objective is to collect spreading debris into an object pile, which is useful for tasks like cleaning.

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
