# OpenReview forum: "D$^3$Fields: Dynamic 3D Descriptor Fields for Zero-Shot Generalizable Rearrangement"
_robot-learning.org/CoRL/2024/Conference — CoRL 2024_

### Official Review · Reviewer_EQkG · 2024-07-04
**very relevant feature representation with evaluation on useful downstream task**

**Originality:** 3
**Technical Quality:** 5
**Clarity Of Presentation:** 4
**Potential Impact:** 4
**Recommendation:** 4
**Confidence:** 4

**Review:**

strengths:
- The paper is well written and easy to follow.
- The proposed work is relevant for many downstream tasks in robotics.
- The features and rearrangement task are evaluated in a real robotic setup in a variety of scenes and objects.

weaknesses:
- The work is directly proposing foundation models as the backbone and only neural fields as an alternative. A short discussion of earlier work on dense feature correspondences, such as Dense Object Nets, would help further to categorise the work and its advantage over previous work.
- Generally, the choice of the foundation models could be better motivated. In lines 74-78, the authors claim that alternative methods that distil features, such as LeRF, "often require dense camera views for a quality field" and that their method "can work with sparse views". This statement is not supported by the paper.
- The paper only shows successful rearrangement tasks and does not discuss failure cases of the proposed approach. The limitations that are discussed mostly stem from the backbones and not from their specific application in the paper.

**Quality Of The Limitations Section:**

2

**Questions For Rebuttal:**

- Why is the depth R interpolated (line 112, 113) when real depth for camera view is available through the RGB-D pair? What exactly is interpolated here? How and why is this done?

- What is the benefit of fusing features in 3D compared to just projecting 2D features from single views to a single-view 3D (a.k.a. 2.5D)? Would a rearrangement task also work with single-view 3D points clouds as long as the objects of interest are in the field-of-view?

- How is the instance mask m_i (eq. 6) obtained from W^p, respectively p_i, and represented? If this is an instance index, then instance indices will not necessarily match between views and cannot simply merged over multiple views. This is broadly discussed in the supplementary material, but the paper is missing information on how to get from p_i to m_i. I suggest briefly mentioning the properties of m_i and p_i, adding an abstract function p_i -> m_i to the paper and referencing pseudo code in the supplementary material on how this mapping is done.

- Why was DINOv2 chosen to provide W^f? Is this solely for its open-set zero-shot performance which does not require collecting new training data? The supplementary material shows additional correspondence accuracy comparisons, such as to Dense Object Nets. However in the paper, the choice of DINOv2 appears mostly arbitrary. Could you better motivate the choice of DINOv2 over alternative foundation model?

- Is there any limitation in how much the starting state and the goal state can differ visually and geometrically? How much do they have to differ before no action sequence between start and target state can be found?

**Robotics Focus:**

4

**Summary Of Paper:**

The paper proposes a feature descriptor that can be queried for arbitrary 3D points in a scene. This feature descriptor is distilled from 2D foundation models such that there is a feature correspondence between a 2D image and a 3D map. This property is evaluated on a downstream task of rearranging a scene to a goal state, given by a target arrangement in a 2D image from a different scene.

**Summary Of Recommendation:**

The proposed implicit feature representation is very relevant to robotics and has been evaluate on a relevant task on a real robotic system. There are only minor issues regarding the choice of DINOv2 left. Post-Rebuttal: The authors addressed my concerns about DINOv2 in additional experiments.

---

### Official Review · Reviewer_vc8A · 2024-07-17
**Needs more details and quantitative results**

**Originality:** 3
**Technical Quality:** 2
**Clarity Of Presentation:** 4
**Potential Impact:** 3
**Recommendation:** 3
**Confidence:** 3

**Review:**

Strengths:
- The paper is well-written, clear, and easy to follow.
- The work demonstrates that one can obtain generalizable feature fields based on multiple RGBD images without having to train a network, which is an interesting concept.

Weaknesses:
- Many system and baseline details are omitted, which make the contributions hard to understand. Additionally, the ablations do not probe the system enough to understand the design choices made for the system. For example, details about how the dynamics model is trained is missing.
- The comparison to the baselines seem rocky. F3RM should be an upper bound as it is not generalizable and instead should provide a highly detailed and crisp feature field. However it is shown to fail in the manuscript and cannot even reasonably reconstruct the scene, which seems odd. More details of the F3RM training could illuminate why it is failing, or rather there may be a bug/design choice that limits its efficacy.
- Most importantly, the main paper appears to have no quantitative results, thus how this work improves the field of generalizable manipulation is completely unknown. I saw 2 subsections in the appendix about correspondense results and some manipulation, but encourage the authors to put this in the main paper so that readers can obtain insights on why the system works well.

**Quality Of The Limitations Section:**

3

**Questions For Rebuttal:**

Questions/Comments:
- The work seems to attack manipulation of unseen objects via generalizable feature fields. The term "zero-shot" is a bit confusing in this context as it appears there is no language being used anywhere.
- Eq 6: how come the weights are not summed to 1?
- Does the dynamics model depend on the point initialisation? How much data is used to train this? How does the dynamics model behave on seen categories vs. unseen categories?
- During manipulation, are the initial tracked keypoints s^0 selected from a single object? If so, which object is selected first and why?
- Line 151: n_s is used for the number of goal points, indicating it is the same number as the sampled points, but the softmax (Eq 8) is applied on all pixels. Is this a typo?
- Where does the virtual reference camera come from? The manuscript says top-down: in this case how are the extrinsics selected? Some goal images are not top-down, including in Fig 1 bottom right (mug) and Fig 6 column 2 (Organize Fruits).
- F3RM is trained with DINOv2. For a more comparison, it should be trained with the same foundation models used for D3 Fields (e.g. Grouding DINO + SAM).
- The F3RM paper reports 50 images from the top hemisphere of the tabletop setup for training the NeRF. Is this the case for the baseline in the paper? I suspect there may only be 4 images used to train the NeRF, which would lead to poor reconstruction results.
- For the debris experiment, what do the masks look like for the initial image and goal image? How does the learned dynamics function behave in this setting? This is quite interesting as granular material dynamics is difficult to predict.
- Section 4.4 does not show anything novel, it is just marching cubes, masks, and features. I'd recommend moving this to the appendix and replacing it with quantitative results.
- Where is X-Mem applied?

**Robotics Focus:**

4

**Summary Of Paper:**

This work proposes to create a neural feature field, called D3 Fields, that leverages several foundation models. The features are computed via heuristic combinations from 4 fixed RGBD viewpoints on a tabletop manipulation environment, thus no learning is needed to compute the feature field. Given a feature field and a goal image, dense correspondences are computed and a simple cost function is optimised via MPC to obtain manipulation actions. Experiments show mainly qualitative results of how D3 Fields obtains better correspondences compared to baselines, and how it can perform rearrangement on unseen objects.

**Summary Of Recommendation:**

POST-REBUTTAL UPDATE: The authors have addressed my main concern of weak baselines that provide no insight to the method. With the new comparisons against a properly trained F3RM, the paper now shows that the instantly computed D3Fields provides qualitative results that are comparable to F3RM (with the appropriate amount of images for training). For completeness, it would be great for the authors to update the manuscript with some quantitative results comparing to F3RM as well. Additionally, there are some ablations that provide more insight into design details of D3Fields. Thus, I have raised my rating to weak accept.

---

### Official Review · Reviewer_FgtV · 2024-07-22
**Review and comments on D3Fields**

**Originality:** 2
**Technical Quality:** 3
**Clarity Of Presentation:** 4
**Potential Impact:** 3
**Recommendation:** 3
**Confidence:** 3

**Review:**

The proposed D3Fields is a concise and effective representation for manipulation task, which mostly follows the first principle. It also compares with classic geometric representation such as pointcloud and more SOTA representation such as feature-based NeRF and show certain advantages over these two types of representations, such as more efficient than NeRF-based representation and more 3D consistent compared to point cloud stitching. The real robot experiments show promising results in terms of generalization ability of goals specified by raw RGB images.

The paper is presented mostly clearly. The experiments are mostly well organized. However, most components of this work are built upon existing works, which might reduce the originality of the proposed method.

Here are some concerns the author might find helpful.

In terms of trajectory optimization in the manipulation tasks, it might be interesting to compare with conventional motion planning methods since the testing scenes and objects are relatively simple. Does dynamics model learning really necessary?

**Quality Of The Limitations Section:**

3

**Questions For Rebuttal:**

The author could consider justifying why learning dynamics model is necessary in this manipulation task.

**Robotics Focus:**

4

**Summary Of Paper:**

This paper presents a 3D representation for manipulation which has the attributes of being 3D, dynamic, and semantic. The multi-view RGBD images are taken as input to build the 3D representation. Vision foundation model features are encoded into the 3D representation. In addition, the dynamics model based on keypoints is also learned to perform trajectory optimization in manipulation. Experiments results of rearrangement tasks show the effectiveness of the proposed methods. The proposed method could also generalize to novel, AI-generated and simulated goal images.

**Summary Of Recommendation:**

This paper proposes a lightweight and effective representation which consists of semantic masks, features from vision foundation models, and dynamics models for manipulation tasks. However, most components are built upon existing work, which might hinder the originality of the contribution.

---

### Author Rebuttal · Authors · 2024-08-10

Our revised main paper and supplement are in the rebuttal file.

---

### Decision · Program_Chairs · 2024-09-04

**Decision:**

Accept

**Comment:**

**Paper summary**

This paper presents a new representation for manipulation-relevant scene features. A key advantage of this method is its performance when using sparse camera views to build a high-quality 3D scene representation. The results show how this representation enables model-based planning to solve rearrangement tasks, without requiring demonstrations like baseline approaches do.

**Review summary**


Summary of strengths:
+ This paper targets manipulation problems that are widespread in robotics and highly relevant to CoRL.
+ The proposed method leverages several pre-existing foundation models, and thus does not require any training to compute the neural field.
+ All reviewers noted that the paper was well-written and easy to follow.


Summary of weaknesses:
- The main paper is missing quantitative results. These are included in the supplementary materials, but should be integrated into the main paper.  **[The revised paper has addressed this concern.]**
- The F3RM baseline does not perform as well as expected, leading to questions about whether it was implemented correctly. **[The revised paper has addressed this concern; the baseline performs poorly when sparse views are used, but performs as expected with more views.]**
- There is limited discussion of when/how the proposed method will fail. **[This concern is not addressed, but is a minor issue.]**

**Response to rebuttal**

The authors have extensively addressed the reviewers' concerns through additional comparisons and visualizations, and restructuring the paper to include quantitative results in the main text (rather than just in the supplementary materials).

Two reviewers maintain their positive ratings (weak accept by FgtV and strong accept by EQkG) and one reviewer has significantly increased their rating (from strong reject to weak accept; vc8A).

**Recommendations for improvement**

* Please follow Reviewer vc8a's suggestion: "For completeness, it would be great for the authors to update the manuscript with some quantitative results comparing to F3RM as well."